# Use, Abuse, and Misuse of Nasal Medications: Real-Life Survey on Community Pharmacist’s Perceptions

**DOI:** 10.3390/jpm13040579

**Published:** 2023-03-26

**Authors:** Elena Russo, Francesco Giombi, Giovanni Paoletti, Enrico Heffler, Giorgio Walter Canonica, Francesca Pirola, Giuseppe Mercante, Giuseppe Spriano, Luca Malvezzi, Enrico Keber, Corrado Giua

**Affiliations:** 1Otorhinolaryngology Head & Neck Surgery Unit, IRCCS Humanitas Research Hospital, Via Manzoni 56, Rozzano, 20089 Milan, Italy; 2Department of Biomedical Sciences, Humanitas University, Via Rita Levi Montalcini 4, Pieve Emanuele, 20090 Milan, Italy; 3Personalized Medicine, Asthma and Allergy, IRCCS Humanitas Research Hospital, Via Manzoni 56, Rozzano, 20089 Milan, Italy; 4Otorhinolaryngology Head & Neck Surgery Unit, Casa di Cura Humanitas San Pio X, Via Francesco Nava 31, 20159 Milan, Italy; 5Società Italiana Farmacia Clinica (SIFAC), Viale Regina Margherita 30, 09124 Cagliari, Italy

**Keywords:** pharmacy, nasal spray, chronic rhinosinusitis, topic steroids, decongestants, allergic rhinitis

## Abstract

Background: Medication overuse is an increasing global problem, especially for those rhinology diseases whose management requires over-the-counter drugs. This observational community pharmacy-based study aimed to investigate the actual use of the best-selling topical nasal medications and to characterize the clinical issues underlying their query through the pharmacist’s perception. Methods: In the pilot phase, a preliminary survey was developed by a team of researchers and tested on a small sample of practitioners to assess usability and intelligibility. Eventual amendments were made according to the feedback obtained, and the final version was submitted to practitioners working in 376 pharmacies evenly distributed over the Italian territory. Results: Two groups of customers (18–30 years old and 60–75 years old) were the ones who most frequently purchased topical decongestants. The dosage applied for sympathomimetic amines was higher than recommended in up to 44.4% and the duration of use longer than 5 days in up to 31.9% of the cases. Patients’ queries of alpha agonists and topical corticosteroids resulted in significantly higher numbers than practitioners’ prescriptions. Allergic rhinitis was the most common disease affecting patients seeking sympathomimetic amines. Conclusions: The prolonged use of sympathomimetic amines in patients suffering from rhinology diseases is a significant problem that requires greater attention in terms of social education and surveillance.

## 1. Introduction

Rhinology diseases, such as rhinitis and chronic rhinosinusitis, may significantly affect a patient’s quality of life and impose a great financial burden on society; therefore, appropriate treatment of these conditions is of utmost importance [1,2,3]. Although the future lies in the development of new drugs capable of targeting specific molecular pathways to prevent long-term adverse effects of systemic therapies and eventually avoid the indication for endoscopic sinus surgery, current topical medications are still the first line of treatment for such conditions. Consequently, patients affected by rhinology diseases often autonomously assume self-medication with over-the-counter drugs and do not respect the proper indications and duration of each treatment [4,5,6,7]. In a large patient survey on allergic rhinitis in France, 44% of patients reported self-medicating frequently [5]. In addition, a study by Mehuys et al. found an overall prevalence of intranasal decongestant overuse of 49%, despite proper education about the limit on the duration of use [8]. Over-the-counter medications, which are generally considered safe, are available without a prescription and can be purchased directly from related pharmacies and stores located in several settings, even on the internet. Therefore, there is a lack of consistent data about the real prevalence of over-the-counter drug misuse, abuse, and dependence. In addition, it should be considered that long-term use of intranasal decongestants has been demonstrated to cause rebound nasal congestion on suspension, thus encouraging further use that can result in hypertrophy of the nasal mucosa associated with other symptoms, such as nasal burning, irritation, and dryness (the so-called rhinitis medicamentosa) [9,10].

Consequently, it is clear that community pharmacists could play an important role in the early detection and prevention of intranasal decongestant abuse by monitoring self-medication and educating patients about the modalities of acute treatments. This has become ever truer since the COVID-19 pandemic. In the era of physical distancing, masks, and personal protective equipment, effective and empathic doctor-patient communication has become increasingly more difficult. Moreover, in some cases, one should not underestimate the role of nonverbal behavior, which, according to Marcinowicz et al., may be perceived in up to 66% of patients in the form of the tone of voice, eye contact, and facial expressions [11]. All these essential aspects of the relationship between patients and physicians have inevitably fallen apart since the advent of social restrictions due to the pandemic, contributing to the emergence of a global healthcare system crisis. In a cross-sectional study of 359 patients attending a tertiary care center in Chennai (India), selected from the outpatient department, wards, and isolation facilities, more than 60% of the participants referred to the difficulty in accessing the health framework and communicating with the doctors. Moreover, in multivariable linear regression analysis, difficulties in communication were shown to have a statistically significant negative impact on trust in doctors [12].

In this context, community pharmacies have been among the few essential services authorized to keep their activity ongoing during the pandemic, thus becoming the most easily accessible healthcare facility to provide advice, information, and drugs to the entire population [13,14]. Accordingly, the need for effective pharmacy programs is becoming more evident. However, accurate data on the patterns of self-medication with intranasal decongestants need to be provided. Thus, this observational community pharmacy-based study aimed to investigate the actual use of some of the best-selling steroid and decongestant nasal sprays. The secondary aim was to characterize the clinical issues underlying the use of these drugs through the pharmacist’s perception.

## 2. Materials and Methods

### 2.1. Study Design

This cross-sectional study is based on an online survey as exempt research and it has been performed in accordance with the ethical standards laid down in the 1964 Declaration of Helsinki and its later amendments. The questionnaire development occurred in two phases. Initially, a team of researchers from the Italian Society of Clinical Pharmacy (SIFAC) and the Department of Biomedical Science of Humanitas University developed a preliminary survey, which was pilot tested on a small sample of practitioners from the SIFAC Group of Clinical Community Pharmacists (SGCP) to evaluate its usability and intelligibility. Then, the final version of the survey was developed according to the feedback obtained during the previous phase.

For the analysis, nasal sprays were divided into 5 different categories according to the pharmacologically active ingredient: sympathomimetic amines, steroids, sympathomimetic amines + antihistamines, isotonic saline solutions, and hypertonic solutions.

The questionnaire consisted of 13 items and was divided into 2 sections as follows:

(a) Geolocation and characterization (urban or rural) of the community pharmacy, education qualification, and years since the graduation (items 1–5)

(b) Characterization of the use and/or abuse of steroid and decongestant nasal sprays (items 6–13). Item 6 investigated age among patients who purchase decongestant nasal sprays. Item 7 investigated how often pharmacists dispense each category of nasal sprays on their initiative. Item 8 investigated the frequency of patients’ requests for each drug category. Item 9 investigated the frequency of sympathomimetic amines use in different rhinology diseases, such as viral rhinitis, allergic rhinitis, nonallergic vasomotor rhinitis, and chronic rhinosinusitis with or without nasal polyps. Item 10 investigated at what dosage sympathomimetic decongestants are used. Item 11 investigated the proportion of patients using nasal sprays at a higher dosage than recommended. Item 12 investigated the average duration of therapy with sympathomimetic decongestants. Finally, item 13 investigated the proportion of patients who use nasal sprays regularly.

The questionnaire was sent electronically to all SIFAC members working in 376 pharmacies evenly distributed over the Italian territory. Data collection took place between 14 March and 22 May 2022 through a Google Form tool and was subsequently analyzed. Each participant signed informed consent for the data analysis.

The survey is available as supplemental content in Appendix B.

### 2.2. Data Analysis

Data were downloaded at the closure of the survey and summarized using descriptive statistics. Categorical variables were classified by count and percentage, while continuous variables were reported as mean ± standard deviation (SD). Finally, ordinal variables were dichotomized as follows: item 6—“sometimes, or less” versus “often, or more”; items 7 and 8—“< 10 times/day” versus “≥10 times/day”; item 9—“≤10 times/day” versus “> 10 times/day”). Difference between paired dichotomous subgroups was tested using chi-square (χ^2^) test.

The sample size was assessed considering a total population of 19,669 licensed Italian pharmacies and a confidence interval (CI) set at 95%. With this sample size, α- and β-error resulted in 0.05 and 0.20, respectively. The statistical power of the study was accordingly set at 80%.

Data analysis was conducted with IBM^®^ SPSS Software for Macintosh, Version 26.0 (IBM Corp., Armonk, NY, USA). Statistical significance was defined as *p* < 0.05.

## 3. Results

### 3.1. General Characteristics of the Study Population

Three-hundred and seventy-six pharmacists working in as many community pharmacies completed the survey. The most represented regions were Lazio (*n* = 66) and Lombardia (*n* = 62), while less represented ones were Valle d’Aosta (*n* = 1) and Umbria (*n* = 1). The study population included 100 (26.6%) rural and 276 (73.4%) urban pharmacies. Most of the participants graduated at least 5–10 years earlier (38.0%), while only a minority of them had more than 20 years of experience (12.0%). One hundred and thirty-two (35.1%) of the included pharmacists had a higher education qualification than a bachelor’s degree. Detailed information about the distribution of the included pharmacies over the Italian territory is shown in Figure 1 and Appendix A, while basic characteristics of the study population are shown in Table 1.

### 3.2. Medication Use

The frequency of decongestant nasal spray purchases stratified by age groups is shown in Figure 2A. Patients who most frequently purchased intranasal decongestants were those between 18 and 75 years old (18–35 years: 26.1% often, 43.9% very often, 5.3% always; 36–60 years: 13.8% often, 44.9% very often, 5.1% always; 61–75 years: 43.1% often, 32.7% very often, 1.3% always). Conversely, requests for medications that resulted in lower percentages were in the outlying age groups (0–17 years: 46.8% often, 5.6% very often, 0.3% always; ≥85 years: 36.4% often, 5.9% very often, 0.0% always). Figure 2B shows the percentages of frequent users for each age group after dichotomizing the frequency categories into “sometimes, or less” versus “often, or more”. Again, in dichotomized analysis, patients between 18 and 75 years old were observed to be the ones who more frequently required topical decongestant nasal spray (61–75 years: 77%; 18–35 years: 75%; 76–84 years: 65%; 36–60 years: 64%; 0–17 years: 53%; ≥85 years: 42%)

Paired data comparison and the corresponding *p*-values are shown in Table 2.

In paired analysis, a statistically significant difference between the two most representative groups of the population (18–35 years old and 61–75 years old) and the other age groups (*p* < 0.05) was observed. Conversely, there was no significant difference between these two groups: 18–35 years old versus 61–75 years old, *p* = 0.549.

The difference between pharmacists’ prescriptions and customers’ queries divided by type of molecules is shown in Figure 3. The drugs most frequently dispensed by pharmacists on their initiative included isotonic and hypertonic saline solutions (≥10 times/day: 82.7% and 81.9%, respectively), while sympathomimetic amines, topical steroids, and sympathomimetic amines combined with antihistamines were less commonly prescribed (≥10 times/day: 69.7%, 58.2%, and 78.2%, respectively). Conversely, the products most requested explicitly by patients were equally distributed among the different categories (≥10 times/day: 84.8% sympathomimetic amines, 84.3% topical steroids, 84.9% sympathomimetic amines + antihistamines, 84.6% saline isotonic solutions, 85.9% marine hypertonic solutions).

The frequency of topical sympathomimetic amines stratified by type of the rhinology disease is shown in Figure 4A. Most patients who purchased topical sympathomimetic amines were those who suffered from allergic rhinitis, followed by those who were affected by viral and vasomotor nonallergic rhinitis (>10 times/day: 27.9%, 22.6%, and 18.1%, respectively). For each disease category, percentages of patients who use sympathomimetic amines more than 10 times/day are shown in Figure 4B. In dichotomized analysis, allergic rhinitis was also observed to be the condition most frequently treated with a high dose of sympathomimetic amines (28%), followed by viral rhinitis (23%), vasomotor nonallergic rhinitis (18%), chronic rhinosinusitis without nasal polyps (14%), and chronic rhinosinusitis with nasal polyps (12%).

Paired data comparison and the corresponding *p*-values are reported in Table 3. There was a statistically significant difference in the frequency of use of topical alpha-adrenergic amines between allergic versus vasomotor rhinitis (*p* = 0.001), as well as between allergic rhinitis and both CRSsNP and CRSwNP (*p* < 0.001). Conversely, a statistically significant difference between the two most representative rhinology conditions (viral rhinitis versus allergic rhinitis, *p* = 0.093) was not observed.

Intake frequency and duration of therapy with intranasal sympathomimetic amines reported by pharmacists are shown in Figure 5A,B, respectively. According to the pharmacists, the dosage applied was higher than recommended (i.e., more than 1–2 puffs for nostril for 2–3 times a day) in up to 44.4% of the cases, while the reported duration of use was longer than 5 days in up to 31.9% of the cases. Moreover, only in 11.9% and 12.5% of the cases, the posology was closer to therapeutic indications and the duration of treatment shorter than 5 days, respectively. Again, according to the respondents, out of 10 patients, more than half exceeded the recommended posology, with a mean of 5.8 ± 2.3. Likewise, on a sample of 10 patients more than half recurrently purchased intranasal sympathomimetic amines (i.e., at least 1 pack every 2 weeks), with a mean of 5.1 ± 2.4.

## 4. Discussion

Self-medication and medication overuse are growing global problems, especially for rhinology diseases. Anti-inflammatory drugs and, above all, corticosteroids, both intranasal and systemic, are at the forefront in the treatment for such conditions, such as EPOS 2020 for chronic rhinosinusitis with or without nasal polyps, as stated in several papers [15,16]. The topical and/or systemic use of glucocorticoids (GC) leads to the suppression of the expression of pro-inflammatory cytokines, chemokines, and adhesion molecules such as ICAM-1 and E-selectin; GC also stimulates the transcription of anti-inflammatory cytokines such as TGF-b. Moreover, GC suppresses pro-fibrotic cytokines related to polyp growth, such as IL-11, the basic fibroblast growth factor (b-FGF), and the vascular endothelial growth factor (VEGF), thus globally reducing the inflammatory load at the level of the nasal mucosa [17]. However, chronic and/or recurrent use of systemic corticosteroids (particularly frequent in patients with CRSwNP and concomitant severe asthma [18]) is associated with a relevant increased risk of developing adverse events (i.e., type-2 diabetes, hypertension, glaucoma, osteoporosis) [19] that may also have a dramatic burden in terms of healthcare costs. Canonica et al., in a recent analysis of the data from the Severe Asthma Network in Italy (SANI) registry, estimated a total annual cost of EUR 242.7 million for severe asthmatic patients due to oral corticosteroid-related adverse events [20]. The amount, which is already impactful, would be even more burdensome if all the conditions manageable with oral corticosteroids, such as chronic rhinosinusitis, were considered.

For these reasons, in order to avoid the chronic assumption of oral corticosteroids, the first line of treatment in rhinology diseases, which are generally characterized by a chronic relapsing course, is based on over-the-counter topical medications, which may be self-administrable in the outpatient setting [4,5,6,7]. Overall, the misuse of over-the-counter drugs is considered more socially acceptable, less stigmatizing, and safer (also due to a likely lack of detection in standard drug screens) than the intake of illicit substances [21]. A systematic review by Schifano et al., although mainly focusing on medications other than topical nasal decongestants (i.e., oral dextromethorphan and diphenhydramine), showed that the misuse of over-the-counter drugs is both globally widespread and popular. As might be expected, vulnerable categories include adolescents and young adults, although real prevalence figures remain unknown [22]. With regard to the awareness of people on the use of nasal decongestants and their possible side effects, a recent observational analysis showed that out of a sample of 385 people only 16.6% of the participants were aware of the side effects, 25.2% of the medically recommended duration of treatment, 21.3% of the possible nasal congestion addiction, and 21.6% of the drugs causing nasal congestion [23].

In addition, since the advent of the COVID-19 pandemic, the need for social distancing and isolation has progressively weakened doctor-patient communication and trust [12,24,25], potentially further fueling the phenomenon of drug misuse and/or overuse. Therefore, the role of community pharmacies in providing advice and information to the entire population is becoming more evident, as well as the need for effective pharmacy programs. Nevertheless, current literature on this topic is scarce and additional data on intranasal decongestants utilization and the patterns of self-medication are required.

To the best of our knowledge, this is the first extensive survey providing data on intranasal decongestant use in Italian community pharmacies. Our results show that customers who most frequently purchased topical decongestants were those between 18 and 75 years old, with 18–30 and 60–75 age groups being the most represented in the overall sample (*p* < 0.05). It is worth noting that, given higher life expectancy, patients in this age group could have a higher risk of long-term adverse effects.

Drugs, most frequently dispensed by pharmacists on their initiative, included isotonic and hypertonic saline solutions, whereas patients’ queries included proportionally higher amounts of sympathomimetic amines, steroids, and antihistamines. Accordingly, it is interesting to observe that a statistically significant difference was seen between queries and prescriptions of alpha-agonists, topical corticosteroids (*p* < 0.001), and alpha-agonists + antihistamines (*p* < 0.05), whereas no significant difference was observed for isotonic and hypertonic solutions. In our survey, patients showed a higher tendency to request active pharmacological principles, mainly sympathomimetic amines and topical corticosteroids, compared with pharmacists who more frequently dispensed non-pharmacological medications.

Moreover, our study showed that customers who most often required topical sympathomimetic amines were those who suffered from allergic rhinitis. This may be surprising considering that the first line of treatment for nasal obstruction in allergic rhinitis is intranasal corticosteroids [26]. A possible explanation may be a patient’s concern about adverse reactions. In a recent study, Hellings et al. [27] found that about 48% of patients suffering from allergic rhinitis are afraid of the side effects of intranasal corticosteroids. Moreover, it has been demonstrated that some patients do not control symptoms satisfactorily despite adequate treatment [28]. Failure to control symptoms leads to a significant impairment of the quality of life with decreased social and work performances. Thus, the lack of efficacy together with the desire for rapid symptom relief may be an additional possible explanation for topical sympathomimetic amine overuse. Finally, according to our survey, the dosage applied for sympathomimetic amines was higher than recommended in up to 44.4% of the cases, whereas the reported duration of use was longer than 5 days in up to 31.9% of the cases.

Alpha-adrenoreceptor agonists act by causing constriction in the capacitance sinusoids of the nasal submucosal vascular layer. Due to their vasoconstrictor action, sympathomimetic decongestants oppose the vasodilation typical of rhinology conditions, reducing nasal airway resistance and thus facilitating nose breathing [29]. The effectiveness of alpha-adrenoreceptor agonists in inflammatory nasal diseases has been clearly established since 1989 when Sperber et al. observed lower total and nasal-specific symptoms in patients assuming pseudoephedrine plus ibuprofen compared with pseudoephedrine alone or placebo, without any statistical significance between the groups in terms of frequency of infections, colds occurrence, and viral shedding [30].

As mentioned above, long-term use of intranasal sympathomimetic amines may cause rebound nasal congestion on cessation, a condition also known as rhinitis medicamentosa [31]. The term ‘‘rebound congestion’’ was used for the first time in 1944 by Feinberg and Friedlaender to describe the nasal congestion experienced after the use of naphazoline [32]. Several pathogenic hypotheses have been formulated to explain this phenomenon:-The number of membrane alpha-adrenergic receptors would be decreased by downregulation due to the chronic assumption of exogenous sympathomimetic amines. Consequently, endogenous noradrenaline production would be decreased by presynaptic negative feedback, thus inducing relative dilatation of the submucosal sinusoid venous plexuses [33];-Adrenergic receptors’ intracellular pathway may become refractory to nasal decongestants, causing the patient to increase the doses of nasal decongestants needed to achieve an effective pharmacological effect, a phenomenon also known as “tachyphylaxis” [34];-The stimulation of adrenergic receptors may induce intense vasoconstriction of submucosal arterioles thus promoting the development of mucosal ischemia and consequent interstitial edema [35].

Even though the exact physiopathology of this condition is still a matter of debate, several researchers have already demonstrated its relationship with the chronic assumption of sympathomimetic amines. In a study conducted on 19 healthy subjects treated with oxymetazoline 200 μg three times daily for 17 days, Vaidyanathan et al. showed a significant reduction in peak inspiratory flow and a nonsignificant increase in inspiratory nasal resistance measured by anterior rhinomanometry compared with the measurements performed in these same subjects before treatment [36].

Similarly, in a study on 18 healthy subjects treated with oxymetazoline 50 μg per day or xylometazoline 280 μg per day for 30 days, Graf et al. demonstrated the presence of mucosal edema on rhinostereometry (optical measurement of the thickness of the mucosa in vivo) after 10 days of treatment, which continued to worsen until the 30th day [37].

In addition to the abovementioned studies, clinical evidence depicts this condition, which may present with inflamed, dry mucosa prone to bleeding, edema, and associated insomnia [31]. These symptoms may prompt patients to use higher quantities of decongestants for a longer period, resulting in both physiologic and psychologic dependence [38]. Thus, it seems crucial to encourage patients to use these drugs properly. In conclusion, emphasis should be given to the relevance of developing preventive measures in the pharmacy setting, such as proper instruction and adequate surveillance of patients.

This study has some limitations, such as the non-randomized inclusion of a limited number of community pharmacies, which increases selection bias and reduces our results’ reproducibility to the whole population. Another limitation is that we used a questionnaire that has yet to be formally validated to investigate the overuse of intranasal decongestants. Finally, customers’ characteristics and medication use were based on self-reporting, which holds the risk of recall bias. Thus, further studies, even randomized, are required to generalize these results to the entire population.

## 5. Conclusions

In the era of personalized medicine, targeted management will become the treatment of choice in such conditions, aimed at a holistic approach to the patient. Genetic, environmental, and behavioral factors, together with the presence of relevant comorbidities should be considered in the multidisciplinary personalized approach in order to reduce the burden of disease on a patient’s quality of life and healthcare-related costs. In this context, our findings highlight that the prolonged and repetitive use of sympathomimetic amines among patients suffering from rhinology disease is a widespread problem that requires greater attention. Possible intervention strategies include the development of preventive measures in the pharmacy setting, such as proper education and adequate surveillance of patients. However, further studies are required to characterize this issue in a larger, randomly selected population sample, as well as in multicentric international settings.

## Figures and Tables

**Figure 1 jpm-13-00579-f001:**
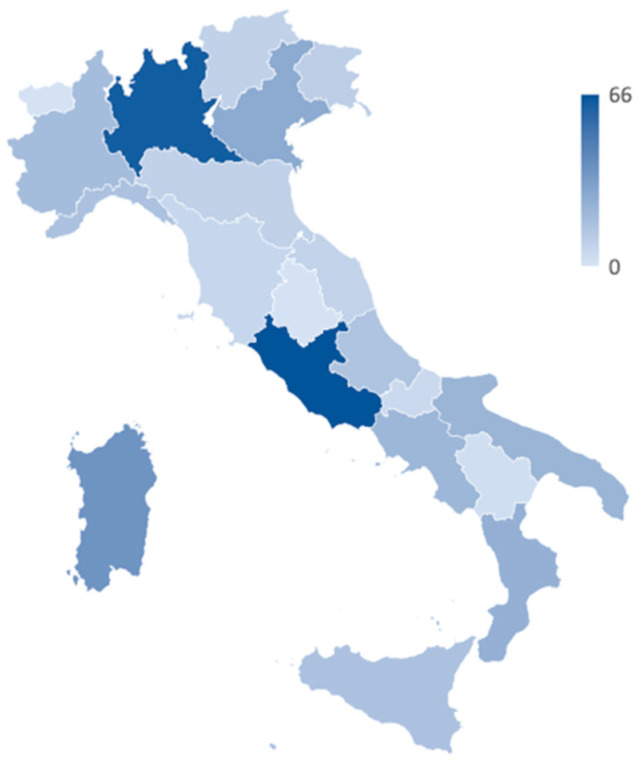
Distribution of the included pharmacies over the Italian territory.

**Figure 2 jpm-13-00579-f002:**
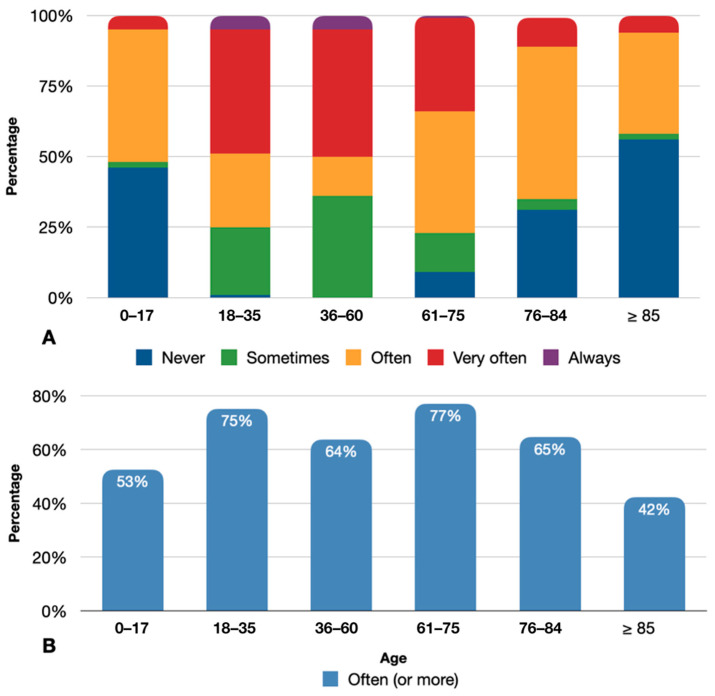
(**A**) Frequency of decongestant nasal spray use stratified by age groups. (**B**) Percentages of frequent users for each age group.

**Figure 3 jpm-13-00579-f003:**
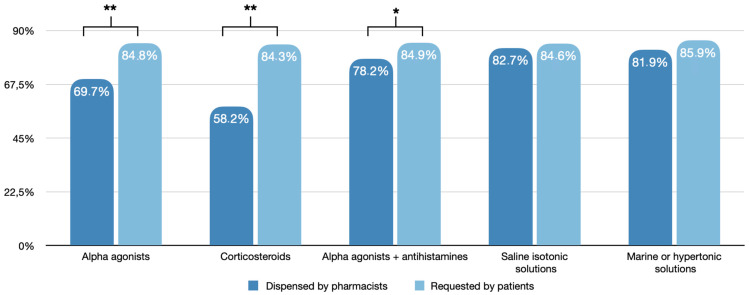
Difference between pharmacists’ prescriptions and customers’ queries of decongestant nasal sprays stratified by type of molecules (* = *p* < 0.05; ** = *p* < 0.01).

**Figure 4 jpm-13-00579-f004:**
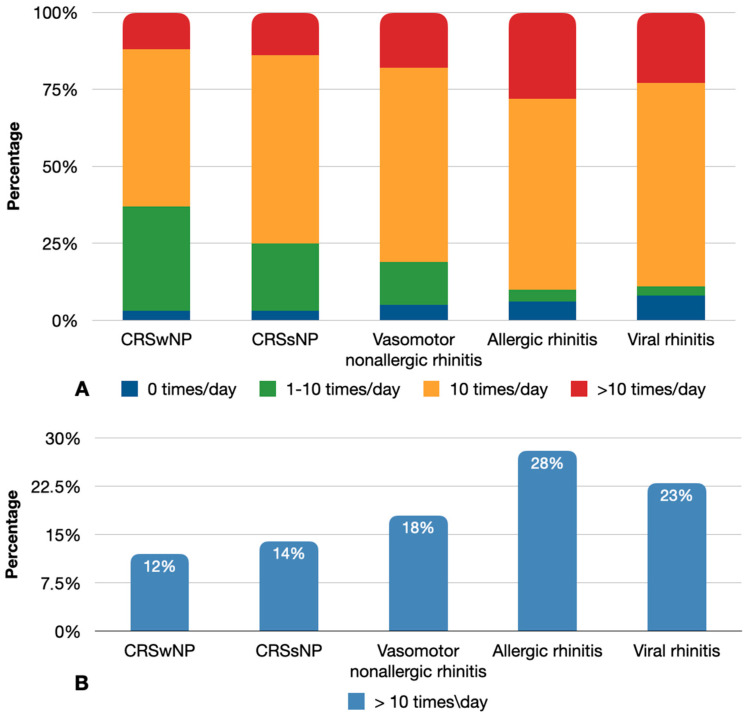
(**A**) Frequency of use of topical sympathomimetic amines stratified by type of rhinology disease. (**B**) Percentages of patients for each disease category who use sympathomimetic amines more than 10 times/day.

**Figure 5 jpm-13-00579-f005:**
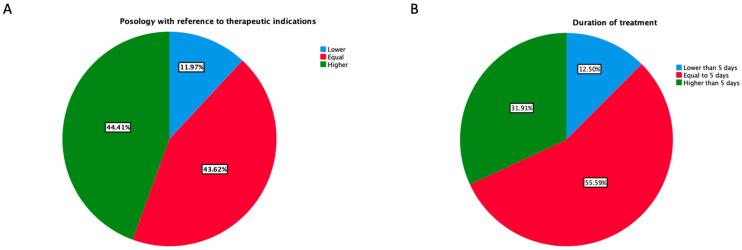
Intake frequency (**A**) and duration of using (**B**) of intranasal sympathomimetic amines reported by pharmacists.

**Table 1 jpm-13-00579-t001:** Characteristics of the population.

		*n*	%
Setting	Rural pharmacy	100	26.6
	Urban pharmacy	276	73.4
Years since graduation	<5 years	88	23.4
	5–10 years	143	38.0
	11–20 years	100	26.6
	>20 years	45	12.0
Educational qualification	Bachelor’s	244	64.9
	Bachelor’s, PhD	3	0.8
	Bachelor’s, Master’s II degree	111	29.5
	Bachelor’s, Master’s II degree, PhD	1	0.3
	Bachelor’s, Master’s II degree, Residency	7	1.9
	Bachelor’s, Residency	10	2.7

**Table 2 jpm-13-00579-t002:** Difference in nasal decongestant consumption by age groups.

	<18	18–35	36–60	61–75	76–84	≥85
<18	-	<0.001	0.002	<0.001	0.001	0.004
18–35	<0.001	-	0.001	0.549	0.002	<0.001
36–60	0.002	0.001	-	<0.001	0.819	<0.001
61–75	<0.001	0.549	<0.001	-	<0.001	<0.001
76–84	0.001	0.002	0.819	<0.001	-	<0.001
≥85	0.004	<0.001	<0.001	<0.001	<0.001	-

**Table 3 jpm-13-00579-t003:** Difference in nasal decongestant consumption by disease. Legend: CRSsNP, chronic rhinosinusitis without nasal polyps; CRSwNP, chronic rhinosinusitis with nasal polyps.

	Viral	Allergic	Vasomotor	CRSsNP	CRSwNP
Viral	-	0.093	0.124	0.001	0.001
Allergic	0.093	-	0.001	<0.001	<0.001
Vasomotor	0.124	0.001	-	0.089	0.025
CRSsNP	0.001	<0.001	0.089	-	0.586
CRSwNP	0.001	<0.001	0.025	0.586	-

## Data Availability

Not applicable.

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
