# Peer review of "Use, Abuse, and Misuse of Nasal Medications: Real-Life Survey on Community Pharmacist’s Perceptions"

_jpm, 2023, doi:10.3390/jpm13040579_

Round 1
Reviewer 1 Report
The frequency and actual use of nasal medications is of great interest to ENT clinicians. This is because it is difficult to evaluate whether it is being used correctly compared to oral medicine. In this study, these questions are properly investigated and conclusions are drawn.
As the author states, I think many clinicians should be particularly well aware that rhinitis medicamentosa is caused by overuse of decongestant nasal spray.
Author Response
Dear Reviewer,
We are pleased by the positive opinion expressed by your comment and hope our research may give a positive contribution to the scientific progress.
Thank you for your consideration. Yours Sincerely
The Authors
Reviewer 2 Report
1. line 41: effective problem-> significant problem
2. line 49: self-medicate-> self-medication
3. line 50: indications and durations -> proper indications and durations
4. line 114: are 19669 Italian pharmacies right?
5. In Table 1: Formation-> Educational Qualification
6. In Figure 3: If the alpha agonists+antihistamine group exists, it is necessary to describe in materials and methods(lines 85-86).
7. In the Table, the description and abbreviation are needed to locate at the bottom.
Author Response
Dear Reviewer,
We are pleased by the positive contribution of your comments to the effective editing of our manuscript. Hereby, we reply to your suggestions poin-by-point
#1 line 41: effective problem-> significant problem
#2 line 49: self-medicate-> self-medication
#3 line 50: indications and durations -> proper indications and durations
According to your suggestions, we adjusted the terminology at line 46 (#1), 58 (#2), 59 (#3).
#4 line 114: are 19669 Italian pharmacies right?
As regards point #4, we confirm that 19669 is the total number of licensed pharmacies on Italian soil, which was considered for the calculation of the minimum sample size for statistical significance.
#5 In Table 1: Formation-> Educational Qualification
We adjusted the row accordingly. Thank you for the effective suggestion.
- In Figure 3: If the alpha agonists+antihistamine group exists, it is necessary to describe in materials and methods(lines 85-86).
The group does exist. Thank you very much for having highlighted this deficiency of the manuscript. Accordingly, we added the category alpha agonists+antihistamine (lines 109-110)
# 7 In the Table, the description and abbreviation are needed to locate at the bottom.
We did modify the layout of the whole tables of the manuscript by shifting the caption at the bottom. Thank you for this advice.
Thank you for your consideration. Yours Sincerely
The Authors